# High Prevalence of Three Potyviruses Infecting Cucurbits in Oklahoma and Phylogenetic Analysis of Cucurbit Aphid-Borne Yellows Virus Isolated from Pumpkins

**DOI:** 10.3390/pathogens10010053

**Published:** 2021-01-08

**Authors:** Vivek Khanal, Harrington Wells, Akhtar Ali

**Affiliations:** Department of Biological Science, The University of Tulsa, Tulsa, OK 74104, USA; vik684@utulsa.edu (V.K.); harrington-wells@utulsa.edu (H.W.)

**Keywords:** survey, cucurbits, potyviruses, CABYV, phylogeny

## Abstract

Field information about viruses infecting crops is fundamental for understanding the severity of the effects they cause in plants. To determine the status of cucurbit viruses, surveys were conducted for three consecutive years (2016–2018) in different agricultural districts of Oklahoma. A total of 1331 leaf samples from >90 fields were randomly collected from both symptomatic and asymptomatic cucurbit plants across 11 counties. All samples were tested with the dot-immunobinding assay (DIBA) against the antisera of 10 known viruses. Samples infected with papaya ringspot virus (PRSV-W), watermelon mosaic virus (WMV), zucchini yellow mosaic virus (ZYMV), and cucurbit aphid-borne-yellows virus (CABYV) were also tested by RT-PCR. Of the 10 viruses, PRSV-W was the most widespread, with an overall prevalence of 59.1%, present in all 11 counties, followed by ZYMV (27.6%), in 10 counties, and WMV (20.7%), in seven counties, while the remaining viruses were present sporadically with low incidence. Approximately 42% of the infected samples were positive, with more than one virus indicating a high proportion of mixed infections. CABYV was detected for the first time in Oklahoma, and the phylogenetic analysis of the first complete genome sequence of a CABYV isolate (BL-4) from the US showed a close relationship with Asian isolates.

## 1. Introduction

The members of the Cucurbitaceae, family, commonly referred to as cucurbits, are major cash crops in the United States (US) and worldwide. Cucurbitaceae includes 15 tribes, 95–97 genera, and approximately 1000 species, which are grown in tropical and sub-tropical regions [1,2,3]. Among them, the genus *Cucumis* (including cucumbers and melons), *Citrullus* (watermelons), and *Cucurbita* (pumpkin sand squashes) are popular. Archeological evidence suggests that pumpkins and squashes (*Cucurbita* spp) were first cultivated in American continents >10,000 years ago; watermelons were first cultivated in Africa at least 4000 years ago, while cucumbers and melons have been in Asia for > 5000 years [4]. In the US, six cucurbits—cantaloupes (*Cucumis melo*), cucumbers (*Cucumis sativus*), honeydew (*Cucumis melo*), pumpkins, squashes (*Cucurbita pepo*, *C. maxima*, and *C. moschata*), and watermelons (*Citrullus lanatus*)—are among the top fifteen fresh vegetables. The value of production from these crops was >two billion US dollars in 2017, which accounts for nearly 13% of the total value of production from fresh vegetables in the US. [5]. In Oklahoma, cucurbits have been cultivated for >70 years, and currently, they are grown across approximately <6000 acres, where viral diseases are one of the major factors limiting their production [6].

Viruses are responsible for causing nearly 50% of emerging diseases in plants [7], which reduce the production of almost all crops and vegetables including cucurbits. More than 59 viruses were reported to infect cucurbit crops in 2012 [8]; however, the number of viruses that could potentially infect cucurbits worldwide has now increased to 96 (Appendix A). In the last two decades, several studies conducted worldwide [9,10,11,12,13,14,15,16,17,18,19] and in the USA [6,20,21,22,23,24] have reported that papaya ringspot virus W strain (PRSV-W), watermelon mosaic virus (WMV), zucchini yellow mosaic virus (ZYMV), cucumber mosaic virus (CMV), cucumber green mottle mosaic virus (CGMMV), and cucurbit aphid borne yellows virus (CABYV) are dominant viruses infecting cucurbits. In the US, >25 viruses infecting cucurbits have been reported [6,7,8,9,10,11,12,13,14,15,16,17,18,19,20,21,22,23,24,25]. The most prevalent viruses infecting cucurbits in the US are PRSV-W, WMV, and ZYMV, which all belong to the family *Potyviridae*. In addition, cucumber mosaic virus (CMV), cucurbit aphid borne yellow virus (CABYV), melon necrotic spot virus (MNSV), squash mosaic virus (SqMV), and cucumber green mottle mosaic virus (CGMMV) have been reported sporadically in cucurbits in different states [6,20,21,22,23,24].

Our previous study [6], conducted during the 2008–2010 growing seasons, was limited to only watermelons in four Oklahoma counties and showed that PRSV-W was the most prevalent virus, followed by WMV and ZYMV, while MNSV, SqMV, and CMV were present with an incidence of less than 1% [6]. The purpose of this study was to further expand our knowledge by surveying more growing areas (11 counties from eight agricultural districts) in Oklahoma (Figure 1) and determine the current status of viruses infecting cucurbits in terms of their relative prevalence and distribution in five major cucurbit crops: cantaloupes, cucumbers, pumpkins, squashes, and watermelons. In addition, a new virus, CABYV, was reported for the first time in Oklahoma. We determined the first complete genome sequence from the US and deduced the evolutionary relationship of CABYV with other complete genomes from around the world.

## 2. Results

### 2.1. Field Symptoms

A wide range of virus symptoms were observed in the cucurbit fields. The most common symptoms were leaf distortion, light and severe mosaics, mottling, and chlorosis (Figure 2A). The other notable symptoms were curling, narrowing, thickening, stunting, rolling, cupping, and a shoestring appearance on the leaves of cucurbits. The ring spots were also observed on pumpkin fruit (Figure 2B). CABYV-positive leaves showed light yellowing and vein thickening on pumpkins (Figure 2C).

### 2.2. Distribution of Viruses in Different Counties

We collected 1331 samples from >90 cucurbit fields during the three growing seasons. Among the 10 tested viruses, PRSV-W was the most dominant virus infecting cucurbits in Oklahoma, with an average infection rate of 59.1%, followed by ZYMV (27.6%) and WMV (20.7%). Apart from these three potyviruses, the other four viruses were present in low abundance: MNSV (5.7%), SqMV (1.4%), CABYV (1.3%), and CGMMV (1.1%). None of the samples tested positive for the remaining three viruses (CMV, cucurbit yellow stunting disorder virus (CYSDV), and squash leaf curl virus (SLCV)). Samples from all eleven counties were infected with at least one virus. The virus prevalence in different counties ranged from 11.8 % (Atoka County) to 100% (Pontotoc County). The leaf samples collected in Atoka County were infected with only one type of virus, while six different viruses were present in the samples from Muskogee County. PRSV-W was present in all eleven counties, followed by ZYMV in 10 counties, WMV in seven counties, MNSV in five counties, and CABYV and SqMV in three counties, while CGMMV was detected in only one county (Table 1).

### 2.3. Distribution of Viruses in Different Hosts

Cantaloupes had the highest viral incidence, with 91.0% of the samples infected with at least one of the 10 viruses tested, followed by pumpkins (86.3%), squashes (84.0%), cucumbers (76.5%), and watermelons (48.2%). None of the three samples from gourds tested positive for any of the ten viruses tested. At least six different viruses were detected in pumpkins, followed by cucumber samples, containing five different viruses, while cantaloupes, squashes, and watermelons were infected with four viruses each. The distribution of PRSV-W was that it was dominant in all five major hosts, and its incidence ranged from 43.0% (133 out of 309) in watermelons to 88.06% (59 out of 67) in cantaloupes (Table 2).

### 2.4. Distribution of Viruses in Different Months and Years of the Growing Seasons

The three potyviruses PRSV-W, WMV, and ZYMV were present in all three growing seasons (2016–2018). PRSV-W was the most dominant virus every year, and its incidence increased each year (Table 3). On the other hand, the incidence of WMV decreased from 40% in 2016 to 7.5% in 2018. The incidence of ZYMV slightly decreased in 2017 from 2016; however, it increased multi-fold in 2018. CGMMV was found only in 2016, MNSV only in 2018, SqMV in 2016 and 2018, and CABYV in 2017 and 2018. The incidence of 10 viruses increased from 66.7% in July to 77.9% in August (Table 4). It marginally decreased in September (71.6%) and increased to as high as 91.5% in October. The incidence of PRSV-W was the highest in all four months, with an average incidence of >46% each month for all three years. Similarly, the average incidence of ZYMV was >20% every month. However, the incidence of WMV increased from 0.6% in July to 36.3% in October.

### 2.5. Distribution of Viruses in Different Growth Stages of Plants

For assessing the viral disease during different growth stages, plants collected during the survey were divided into three groups, namely, the pre-flowering stage (i.e., plants before flowering), fruiting stage (i.e., plants from flowering to having immature fruits), and harvesting stage (i.e., plants bearing fruits ready to harvest and beyond). The incidence of viruses in the pre-flowering stage of the plants was 55.6%, followed by the fruiting stage with 67.9%, and by the harvesting stage, the virus incidence reached 85.1%. The incidence of PRSV-W during different growth stages did not differ significantly; however, the incidence of ZYMV gradually increased from 20.3% at the pre-flowering stage to 29.6% at the harvesting stage. Interestingly, the incidence of WMV was zero at the pre-flowering stage but overtook ZYMV at the harvesting stage, with an incidence of 30.6% (Table 5).

### 2.6. Mixed Infections

In this study, 1023 out of 1331 (76.9%) symptomatic cucurbit samples tested positive for at least one of the 10 tested viruses. Out of the 1023 virus-positive samples, 434 samples were infected with more than one virus, which accounts for 32.6% of the total samples collected. There were as many as 21 different combinations of mixed infections from the field samples based on the dot-immunobinding assay (DIBA) results (Table 6). A total of 343 samples were infected with two different viruses, 84 were infected with three different viruses, and seven samples were infected with four different viruses. The most common mode of double infection was PRSV-W+ZYMV, followed by PRSV-W+WMV and WMV+ZYMV. A combination of MNSV, PRSV-W, and ZYMV was the most common triple infection, followed by the combination involving three potyviruses: PRSV-W, WMV, and ZYMV. Three different combinations of four viruses infecting the same samples were observed. Notably, PRSV-W, WMV, and ZYMV were common in all the quadruple mixed-infection combinations (Table 6).

### 2.7. Statistical Analysis of Various Virus Infections

There were 76 sample populations, of which 14 were eliminated from the statistical analysis because they had less than six samples. All samples were classified as either infected or uninfected by any virus, with H_o_ signifying that there was no difference among the 62 sample populations in the distribution of plants in the two categories. The infected samples were not split into those with a single virus versus multiple viruses due to the low expected number of multiple-virus-infected plants in many sample populations. There was a significant difference among sample populations (*X*^2^ = 841.08, df = 61, *p* < 0.0001). In order to determine why the significant differences among sample populations existed, the virus distributions were analyzed among the three years of the study (2016–2018), hosts, growth stages, or counties from where they were collected.

The sample populations from each year of the survey were analyzed separately for the overall infection rate. All three tests corresponding to three sample-collection years revealed significant differences among the particular year’s sample populations (2016: *X*^2^ = 239.83, df = 15, *p* < 0.0001; 2017: *X*^2^ = 242.19, df = 9, *p* < 0.0001; *X*^2^ = 282.85, df = 35, *p* < 0.0001), which indicated that yearly differences did not explain the variation in the overall infection rate. The samples from all five hosts were classified by developmental stage, and within each classification, each sample was either uninfected, infected by a single virus, or infected by two or more viruses. For cantaloupes, cucumbers, and squashes, there were no samples collected during the flowering stage. Each host was tested separately for the independence of the developmental stage and infection category (H_o_ = independent). For all five hosts except cantaloupes, a significant difference from independent association of developmental stage and infection category existed (cantaloupe: *X*^2^ = 1.44, df = 2, *p* = 0.4873; cucumber: *X*^2^ = 6.77, df = 2, *p* = 0.0340; pumpkin: *X*^2^ = 51.20, df = 4, *p* < 0.0001; squash: *X*^2^ = 13.30, df = 2, *p* = 0.0013; watermelon: *X*^2^ = 63.68, df = 4, *p* < 0.0001). The cantaloupes had only six samples in the fruiting stage, which may have accounted for the non-significant result. Similarly, the virus incidence was also analyzed as a product of the spatial location of the hosts by grouping sample populations solely based on the county from where they were collected without regard to the date collected, plant species, or developmental stage. There were eleven groups corresponding to the eleven counties of this study. The samples fell into one of three categories: uninfected, infected by a single virus, or infected by two or more viruses. There was a significant difference in the distribution of viruses in the three categories from different counties (*X*^2^ = 304.90, df = 20, *p* < 0.0001). Finally, the predicted numbers against observed numbers generated from the multinomial target theory approach for each year were tested separately to test the possibility of random independent events. Only virus types present in a particular year were used to calculate probabilities. The data from 2016 (*X*^2^ = 77.6, df = 1, *p* < 0.001) and 2018 (*X*^2^ = 42.4, df = 1, *p* < 0.001) differed significantly from the independent event model. By contrast, the 2017 data did not differ from the model predictions (*X*^2^ = 1.4, df = 1, *p* = 0.237).

### 2.8. Molecular Confirmation of Selected Viruses

More than 10% of the DIBA-positive samples for PRSV-W, WMV, and ZYMV were further confirmed at the molecular level with RT-PCR using specific primers. The expected PCR products for PRSV-W, WMV, and ZYMV, respectively, were 963 bp, 979 bp, and 902 bp (Figure 3). After sequencing the PCR products, the lengths of the CP gene of the three viruses were 864, 852, and 840 nucleotides (nt) for PRSV-W, WMV, and ZYMV, respectively. One isolate each of ZYMV and CABYV was also completely sequenced [26,27]. None of the selected DIBA-negative samples were positive according to RT-PCR using primer pairs for the aforementioned viruses.

### 2.9. Sequence Analysis of CABYV BL-4 Isolate

The complete genome of the CABYV BL-4 isolate was 5679 nt long and was the first complete genome sequenced from the US [26]. The schematic representation of the genomic organization of the BL-4 isolate is illustrated in Figure 4. The genome contained a 20-nt-long 5’ untranslated region (UTR) and 165-nt-long 3′ UTR. The open reading frame (ORF) contained six overlapping proteins, with a 199-nt-long intergenic non-coding region (NCR) between the P1–P2 fusion protein and readthrough protein (P5). The P1–P2 protein has ribosomal slippage at nt position 1488, and P5 has a stop codon at nt position 4108–4110 (at the end of the coat protein region). The nt positions of six different proteins and their respective amino acid lengths are shown in Table 7. The nt similarity of the CABYV BL-4 isolate was >90% with Chinese, Korean, and Japanese isolates and <90% with other CABYV isolates (Table 8). The query cover for the Brazilian isolates was only 61% because it is a recombinant strain with around 60% CABYV at the 5’ end and 40% of unknown virus origin at the 3’ end of the genome [28].

### 2.10. Phylogenetic Analysis of CABYV Isolates

The maximum likelihood (ML) tree inferred in MEGA7 [29] using default parameters revealed two major (G1 and G2) phylogroups (Figure 5). The G1 phylogroup contains CABYV isolates from Europe and Brazil, while the G2 phylogroup contains isolates from the rest of the world. The G1 phylogroup is further divided into two subgroups (G1a and G1b). G1a contains European isolates and two Brazilian isolates form a separate cluster (G1b). Similarly, the second phylogroup is divided into three subgroups: first, the BL-4 isolate together with Chinese, Korean, and Japanese isolates (G2a); second, the Asian recombinant strains from China and Taiwan (G2b); and third, the non-recombinant isolate from Taiwan together with PNG, East Timor, and Indonesian isolates (G2c). The Bayesian inference (BI) tree deduced from Bayesian evolutionary analysis sampling trees (BEAST) also showed two distinct phylogroups as in the ML tree (Appendix A). While the clustering pattern within the first group was identical to the ML tree, the second group has two additional subgroups: first, the US isolate with the Beijing isolate, and second, the CABYV-C isolate from Taiwan, thereby making five subgroups.

### 2.11. Recombination Analysis of CABYV Isolates

Recombination analysis showed that a total of 61 possible recombination events were detected by all seven algorithms within Recombination Detection Program version 4 (RDP4) [30]. Only the recombination events detected by at least three methods with high statistical support were considered. Altogether, there were 22 recombination events involving 24 isolates. The BL-4 isolate from the US is a major parental isolate for the HD118 and HD1 isolates from Korea, with recombination detected at nt position 854–3862 (Figure 6). While all the methods except GENECONV identified BL-4 as a parental isolate of HD118, only four methods—RDP, Bootscan, MaxChi, and 3Seq—identified it as a major parental isolate of HD1. Similarly, the BL-4 isolate also served as a minor parent for the C-TW2 isolate from Taiwan at the 5′ end of the genome (Table 9).

## 3. Discussion

This study presented three years (2016–2018) of survey data for 10 important viruses infecting major cucurbit crops in Oklahoma. The survey results showed that viruses are common in cucurbits, and potyviruses are a major and continuous threat to cucurbit production in Oklahoma.

The symptoms appearing on virus-infected plants afflicted by the three potyviruses were similar, so it was difficult to distinguish the virus type in the field based on appearance, except for ringspot caused by PRSV-W on fruits. Around 77% of those symptomatic plants sampled were positive for at least one of the tested viruses. The remaining samples might have been infected with viruses that were not tested in this study. The symptoms caused by viruses in pumpkins, squashes, and cantaloupes were more severe than those caused by virus infections in cucumbers and watermelons.

Our previous study [6] only focused on watermelon plants and was conducted in four Oklahoma counties, which showed that PRSV-W was the most dominant virus, followed by WMV and ZYMV. This study shows that the prevalence among the three potyviruses has changed over time. For example, PRSV has maintained its top ranking, while ZYMV has overtaken WMV in the overall incidence. CABYV was reported for the first time in Oklahoma during this study [31], which presents the first complete genome of CABYV in the US. In addition, the DIBA results suggest that 15 cucumber samples out of 19 collected from Cimarron County in 2016 were infected with CGMMV. CGMMV was recently reported in the US [23], and this is the first report of it from Oklahoma. Since CGMMV is seed transmitted, it could pose a problem in other states unless preventive measures are applied. The incidence of MNSV and SqMV reported in this study was low and was similar to that found in the previous study [6]. Interestingly, there was no CMV infection during the 2016–2018 growing seasons. The prevalence of CMV was less than 1% in Oklahoma during the 2008–2010 survey [6], and there were zero incidences in this study. These results suggest that despite being one of the most widely distributed viruses infecting plants, including cucurbits [32], CMV might have been eliminated from the major cucurbits in Oklahoma due to the absence of virus inoculum in alternate hosts and vectors. The primary mode of virus transmission (Table 10) for four (three potyviruses and CABYV) of the seven viruses detected in this study is aphid vectors, which accounted for >90% of total virus infections. Three viruses (ZYMV, MNSV, and CGMMV) are also transmitted by infected seeds. The cultivation of different cucurbit crops in near proximity was also frequently observed during the course of this study, which might have enabled viruses to spread among other cucurbit hosts easily. These results suggest that controlling aphid populations early in the growing season, the use of virus-free seed, and good agricultural practices are key in minimizing viral epidemics in cucurbit crops in the state.

Based on our results, the dynamics of viruses differ among the various cucurbit hosts. Altogether, four viruses (MNSV, PRSV-W, WMV, and ZYMV) were present in all five major cucurbit crops. CGMMV was only present in cucumbers, and CABYV and SqMV were only present in pumpkins. PRSV-W was dominant in each host, and the distribution of other viruses was significantly different in different hosts. For instance, MNSV was present in 74.6% (50 out of 67) of the samples from cantaloupes but was present with low incidence in other hosts, such as 39.2% (20 out of 51) in cucumbers, 2.6% (4 out of 156) in squashes, and less than 1% incidence in watermelons (1 out of 309) and pumpkins (1 out of 745). Similarly, the incidence of WMV was higher in pumpkins, squashes, and watermelons than cucumbers and cantaloupes (Table 5). ZYMV was more prevalent than WMV in cucumbers, cantaloupes, and squashes, but WMV was frequently found in pumpkins and watermelons. These host preferences of the different viruses among the various hosts could be driven by the preferences of their vectors for particular hosts present in the respective fields or nearby alternate hosts. Although the composition of the viruses in different years changed, PRSV-W, WMV, and ZYMV were consistently present, albeit with different rates of incidence. The other four viruses were mostly sporadic; however, outbreaks of these viruses in the near future cannot be ruled out. CABYV requires more attention owing to its rapid emergence and high incidence in Europe and Asia [12,13,33,34].

In 2016, ZYMV was not detected in Caddo, Muskogee, McCurtain, Payne, and Tulsa counties. However, in 2018, the virus was detected in all these locations. Moreover, it was detected in additional counties we surveyed in 2018 including Cherokee, Carter, and Pontotoc. These results suggest that ZYMV is an emerging virus infecting cucurbits in Oklahoma. Although there are only a few alternative hosts that are potential reservoirs or overwintering hosts for ZYMV [35], it was found to be highly stable and could be transmitted via contact-mediated wounding and/or abrasion as well [36]. Unlike PRSV-W and WMV, ZYMV can spread even when aphids are absent because ZYMV can be transmitted via virus-infected seeds [37]. ZYMV-infected seeds might have provided primary infection foci for the virus, and the other two modes of transmission (aphids and contact) helped in the transmission of ZYMV to adjacent plants. We also speculate that the rapid spread of the virus to other locations might be due to its ability to be transmitted by seeds. Interestingly, while the prevalence of WMV and ZYMV was significantly higher at the harvesting stage than at the young (pre-flowering and fruiting) stage, the incidence of PRSV remained similar throughout the different developmental stages.

The viral infection rates varied significantly among different counties, hosts, collection years, and growth stages. The discrepancy in the infection rates among different counties and collection years might be due to the presence or absence of wild hosts for these viruses and simply due to differences in agricultural practices such as crop rotation and the use of virus-susceptible varieties. The relatively higher infection rates in the four hosts other than watermelons could also be attributed to the larger sizes of the leaves, which would facilitate the aphid trapping. A gradual increase in the infection rate was observed as the plants grew older. After the flowering stages of plants, the plants are more likely to attract aphids, which in turn increase the chances of the transmission of viruses from infected plants to uninfected plants nearby. In addition, as the plants grow larger and cover more area, transmission to and from other plants becomes more likely. It was previously reported that aphid performance and movement are correlated with the age and species of plants [38]. Therefore, a higher incidence of viruses with the progression of plant growth was also probably due to the higher activity of aphids during the later stages of plant growth. Another factor, which could result in increased viral infection during later stages of the host, is physiological changes induced by the virus in infected plants that result in attracting vectors to infected plants in preference to healthy plants. For instance, ZYMV helped to increase the production of volatile compounds in plants, facilitating the attraction of a higher number of aphid vectors [39]. Similarly, a PRSV-W and *Aphis gossypii* combination resulted in an increase in the vector population by plant enrichment, thereby allowing the swift transmission of the virus in the same field or to nearby fields [40]. In another study, aphids acquiring CMV were found to encourage the virus to move towards uninfected plants by reducing the quality of the vector’s host plants [41].

A high proportion of mixed virus infections (434 out of 1023 samples, 42.42%) was observed among the infected samples. The rate of mixed infections involving three potyviruses was 88.94% (386 out of 434) of the total mixed infections recorded in this study and 37.73% of the total infected samples (Table 6). Our previous study [20] revealed that the most common combination of mixed infections in the southern United States was PRSV-W+WMV, followed by WMV+ZYMV and PRSV-W+ZYMV. In Oklahoma, 18.9% of samples were found to be infected with at least two viruses among 869 cucurbit samples [6]. A high incidence of mixed infections involving PRSV-W, WMV, and ZYMV has also been reported previously in different studies around the world: in the USA [6,20,22], Iran [12], Spain [13], Brazil [42], Panama [14], Venezuela [43], South Africa [44], Ivory Coast [45], Argentina [18], India [16], and France [19]. However, the proportion of mixed infections involving these three viruses in this study was higher than that in previous studies.

Mixed infections lead to not only symptom variation but also a change in the pattern of vector transmission and infectivity. In addition, the preference and fitness of vectors can also be changed differently by mixed virus infections compared to single infections [46]. They may also lead to virus–virus interactions within a host, resulting in new variants with novel features [47]. Mixed virus infections can be antagonistic [47,48,49,50], synergistic [42,51,52,53], or neutral [53,54]. Additionally, the suppression of post-transcriptional expression by one virus helps in the replication of another virus, thereby forming a synergistic association [55]. Sometimes, the same plant can be co-infected with a different virus strain, which leads to cross-protection [56]. The cross-protection strategy has been used to control different viral species including potyviruses [57]. At this stage, it is not clear whether the mixed infections in this study were synergistic or not. The symptoms shown by the virus-infected cucurbit plants in mixed infections were more severe than in single infections, indicating the possibility of synergism. However, the gradual increase in ZYMV and the reverse in WMV seen in our survey from 2016 to 2018 raised suspicion regarding synergism. A recent study [39] conducted on mixed infections involving WMV and ZYMV in a greenhouse setting showed that ZYMV dominated WMV. ZYMV replicated at a similar rate in both single and mixed infections, whereas WMV replication sharply declined in combination with ZYMV due to in-plant competition. However, the close evaluation of the interaction of these viruses in the field is desired to reach a conclusion about the synergism and competition. Additionally, the role of another dominant virus, PRSV-W, should be determined in future studies.

There have been limited studies about the incidence of viruses infecting cucurbits lately, but this study indicates that the distribution of viruses in cucurbits changes over time. The vector population, the inoculum source, the susceptibility of cultivars to a virus, and environmental factors play an important role in the inconsistency of virus incidence at different time points and locations.

Despite being geographically distant, the CABYV BL-4 isolate was found to be evolutionarily closer to Asian isolates. Based on evolutionary analyses, CABYV was broadly classified in the Asian and Mediterranean subgroups by Shang et al., 2009 [58], and in four groups (Asian, Mediterranean, Taiwanese, and recombinant) by Kwak et al., 2018 [34]. The phylogenetic grouping of the isolates was loosely based on geographical location in both studies. Our study shows a similar result to the 2009 study [36] but with modified subgrouping due to the addition of other isolates from different parts of the world. The Asian group has expanded to include the US and PNG isolate as well (Figure 5). Similarly, the European group clusters together with Brazilian isolates, with high bootstrap and posterior probability values in ML and BI phylogenetic trees. Hence, we propose the first group (G1) as the European and Brazilian isolates and the second group (G2) as the Asian and US isolates (Figure 5).

In RNA viruses, including *Luteoviridae*, recombination is considered a main evolutionary force that shapes their diversity. The intergenic noncoding region (IR) of the Luteoviruses, in particular, is considered a hotspot for genome recombination [59,60]. This study further corroborates this evidence, as 11 out of 22 possible recombination events involved this region. Although none of the recombination detection methods in RDP4 found any evidence of the US isolate being a recombinant, it served as a major parental isolate for two Korean isolates: HD1 and HD118. The BL-4 isolate also serves as a minor parental isolate for the Taiwanese isolate C-TW2, with Sq-2003 being the major parent. Interestingly, the HD1 and HD118 isolates were detected as products of recombination between HS2 from Korea and CZ from China in a recent study [34]. The nucleotide positions of the recombination breakpoints detected for these isolates in this study were also different. The discrepancy might be due to the enhanced robustness in the analysis provided by the inclusion of 12 additional complete genomes in this study. In addition, the overrepresentation of South Korean isolates and the smaller number of isolates from other countries might have caused inconsistencies in the recombination and phylogenetic analysis.

In conclusion, our study has provided further knowledge about and meaningful insights into the epidemiology of various viruses and the fluctuation in the distribution and prevalence of these viruses over time. In addition, the re-emergence of known or emergence of newly reported viruses will also be important for future management studies for a particular crop and locality.

## 4. Materials and Methods

### 4.1. Survey Area and Sampling

Surveys of viruses in cucurbit crops were conducted during three growing seasons (2016–2018) in eight out of nine agricultural districts of Oklahoma (Figure 1), including one to three counties in each district. A total of 11 counties that include Cimarron (panhandle district), Blaine (west-central), Caddo (south-west), Payne (central), Carter, Atoka, Pontotoc (south-central), Tulsa (north-east), Muskogee, Cherokee (east-central), and McCurtain (south-east) were surveyed. Overall, there were more than 90 fields, ranging from those of a few acres with small-scale farms to large commercial fields up to 50 acres in area.

The six cucurbit hosts (cantaloupes, cucumbers, gourds, pumpkins, squashes, and watermelons), depending on their availability in the respective counties, were sampled. Young, fully expanded leaves showing typical virus-like symptoms (symptomatic) and some healthy leaves (asymptomatic) were randomly collected from different cucurbit hosts. The number of leaves sampled from each plant ranged from 2 to 4. The collected leaves were put in individual Ziploc plastic bags, labeled, brought to the laboratory on ice, and processed within 48 h of collection.

### 4.2. Sample Preparation for Dot-Immunobinding Assay

Samples were prepared and tested with the dot-immunobinding assay (DIBA) as described previously [6,61,62]. Approximately 100 mg of leaf tissue from each sample was crushed in an individual plastic bag containing one volume of phosphate-buffered saline (PBS), pH 7.4, at room temperature. The extracted sap was then centrifuged for 2–3 min to obtain a clear supernatant, and 2 µL of the supernatant was dotted and replicated on 10 nitrocellulose membranes (Bio-Rad laboratories, Hercules, CA, USA). Each nitrocellulose membrane was made to accommodate 121 samples (11 × 11). Leaf sap extracted from healthy squashes grown in the growth chamber was used as a negative control, while positive controls for each virus were obtained commercially (Nano Diagnostics, LLC, Fayettville, AR, USA and Agdia, Inc, Elkhart, IN, USA) and were also dotted on each membrane. After dotting, the membranes were air-dried for 10 min and stored at 4 °C until development.

During the membranes’ development, all the blocking and antibody solutions were prepared in PBS, pH 7.4, and contained 600 mM glucose and 600 mM mannose. All incubations were performed at room temperature with gentle agitation on a rocking roller. After the incubation of the membranes with their respective virus-specific polyclonal antibodies, all the membranes were washed three times (10–15 min/wash) in in AP 7.5 (alkaline phosphatase buffer—100 mM Tris-HCl, pH 7.5, 100 mM NaCl, 2 mM MgCl_2_, and 0.05% Triton X-100)—for 20 min each. After washing, the membranes were incubated in PBS for 1 h with a goat anti-rabbit IgG-alkaline phosphatase conjugate antibody (Southern Biotechnology, Birmingham, AL, USA). The membranes were washed again as described above. Before adding the substrate, all the membranes were briefly washed once in AP 9.5 (alkaline phosphatase buffer: 100 mM Tris-HCl, pH 9.5, 100 mM NaCl, and 5 mM MgCl_2_). The membranes were incubated in AP 9.5 containing 0.33 mg of nitro blue tetrazolium (NBT) per mL and 0.17 mg of 5- bromo-4-chloro-3-indolyl phosphate (BCIP) per mL (VWR, USA) in the dark for 5 to 10 min. The reaction was stopped with 10 mM Tris-HCl, pH 7.5, containing 5 mM EDTA. Both positive and negative results were determined visually according to the intensity of the color in comparison with the positive and negative controls.

All samples dotted on replicated membranes were tested against the polyclonal antisera of 10 viruses, including CGMMV, CMV, CABYV, CYSDV, MNSV, PRSV-W, SLCV, SqMV, WMV, and ZYMV (Table 10).

### 4.3. Statistical Analysis

Chi-square tests were used to determine if differences existed in the infection rates among the sample populations from all three years. An infected plant was defined as harboring any of the 10 viruses tested. The sample populations were based on the date of collection (month/year), location (counties), plant host (cantaloupes, cucumbers, pumpkins, squashes, or watermelons), and developmental stage (pre-flowering, fruiting, or harvesting). Samples from gourds were not included in the analysis due to the low sample population size (*n* = 3).

The differences in incidence rates among the sample populations were tested for each year, and different developmental stages within each plant species were analyzed. Spatial effects on the county level were also analyzed. For these analyses, the entire data set was used since sample populations were being combined based on the year of collection, species, or county. In each analysis, chi-square statistical tests were used.

Finally, we tested whether the infection of plants by multiple virus types could be explained as the result of random independent events. The multinomial target theory approach was used to generate the expected number, which was used in a chi-square test of whether the phenomenon of multiple virus infections was due to random, independent infection events. For each virus species, the probability that a plant would be infected was calculated by using the observed data without accounting for whether a plant was or was not infected with multiple viruses. These values were used to deduce the probabilities for each of the single infections (i.e., P_1_*(1−P_2_)*(1−P_3_)*…, the probability of no infection (i.e., (1−P_1_)*(1−P_2_)*(1−P_3_)*…) and the probability of multiple virus infections (i.e., at least two infections, 1−P (no infection)–Σ P (infected only by virus i)).

### 4.4. Molecular Confirmation of Selected Viruses

The confirmation of the three potyviruses (PRSV-W, WMV, and ZYMV) was performed by reverse transcription–polymerase chain reaction (RT-PCR) in at least 10% of the DIBA-positive samples. For WMV, primers from the previous study [6] were used for amplifying the coat protein (CP) gene. For amplifying the PRSV and ZYMV CP genes, primers were designed using the primer3 software package available online, and the following primers were used (PRSVCPF: 5′-CTGATGATTATCAACTTGTT-3′; PRSVCPR: 5′-TAAGGTGAAACAGGGTGGAG-3′; ZYMVCPF: 5′-GAACAAGGAGACACTGTGAT-3′; ZYMVCPR: 5′-GCAGCGAAACAATAACCTAG-3′). The expected PCR-amplified product sizes of 963, 979, and 902 for PRSV-W, WMV, and ZYMV, respectively, included a part of the Nib gene, the full CP gene, and a part of the 3′ untranslated region. A small portion (50–100 mg) of the infected leaf was used to extract total RNA by the TRI Reagent (Molecular Research Centre Inc, Cincinnati, OH, USA) method as described previously [6]. Five microliters of the RNA template was used for cDNA synthesis, and 1 µL of the cDNA was used for the PCR. The PCR conditions for ZYMV and PRSV were 94 °C for 2 min for initial denaturation, and then 30 cycles of 94 °C for 30 s, 50 °C for 40 s, and 72 °C for 40 s, with a final extension at 72 °C for 10 min. For WMV, the annealing temperature was 48 °C. The amplified PCR products were run in a 1% agarose gel, and expectedly sized DNA fragments were excised and gel purified (Qiagen, Hilden, Germany). Three microliters of each gel-purified product was ligated using the pGEM-T Easy Vector (Promega Corp, Madison, WI, USA) and transformed into *Escherichia coli* DH5α competent cells (New England Biolabs, Ipswich, MA, USA). Transformed cells were subjected to blue–white screening using Luria–Bertani agar (LBA), carbenicillin, isopropyl-thiogalactopyranoside (IPTG), and X-gal. Three to five clones were selected and sequenced in both directions by Sanger sequencing using an Applied Biosystems 3130 at the Department of Biological Science, the University of Tulsa, Oklahoma. Nucleotide sequences were analyzed using the basic local alignment search tool (BLAST).

### 4.5. Phylogenetic Analysis of CABYV Complete Genomes

A CABYV BL-4 isolate was collected during the 2017 growing season from Blaine County in Oklahoma and was completely sequenced as described previously [26]. The 45 published complete genome sequences of CABYV (Appendix A) were retrieved from GenBank (ncbi.nlm.nih.gov). These nucleotide sequences were aligned using ClustalX in MEGA7. The ML and BI trees were built with default parameters in MEGA7 and BEAST, respectively. Both trees were visualized in Figure version 1.4.3.

### 4.6. Recombination Detection

Recombination events, putative minor and major parents of recombinant isolates, and recombination breakpoints were analyzed using seven methods incorporated in RDP4 with default settings for all 45 complete genome sequences available in the GenBank database. Only recombination events with Bonferroni-corrected P-values less than the cut-off of 0.05 in three or more methods were regarded as recombinants to reduce the false detection of recombination. The recombination events that were indicated by RDP4 as “may be due to factor other than recombination” were also excluded. The breakpoint locations and recombination region were verified individually by using methods that detected the recombination.

## Figures and Tables

**Figure 1 pathogens-10-00053-f001:**
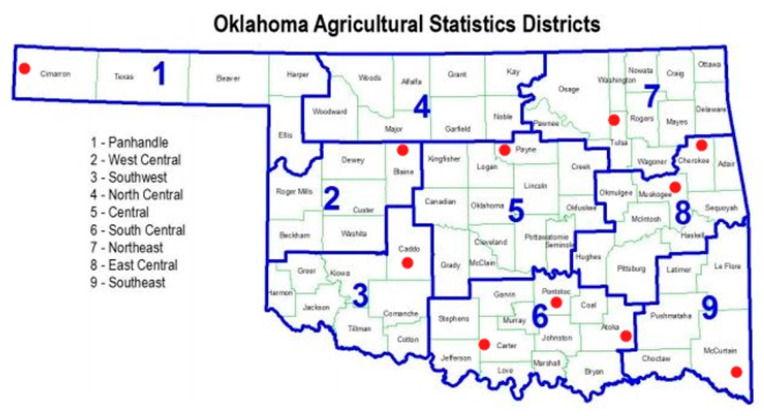
Geographical locations of different counties of Oklahoma. The solid blue lines denote borders of different agricultural districts, and the red dots represent the counties from where samples were collected.

**Figure 2 pathogens-10-00053-f002:**
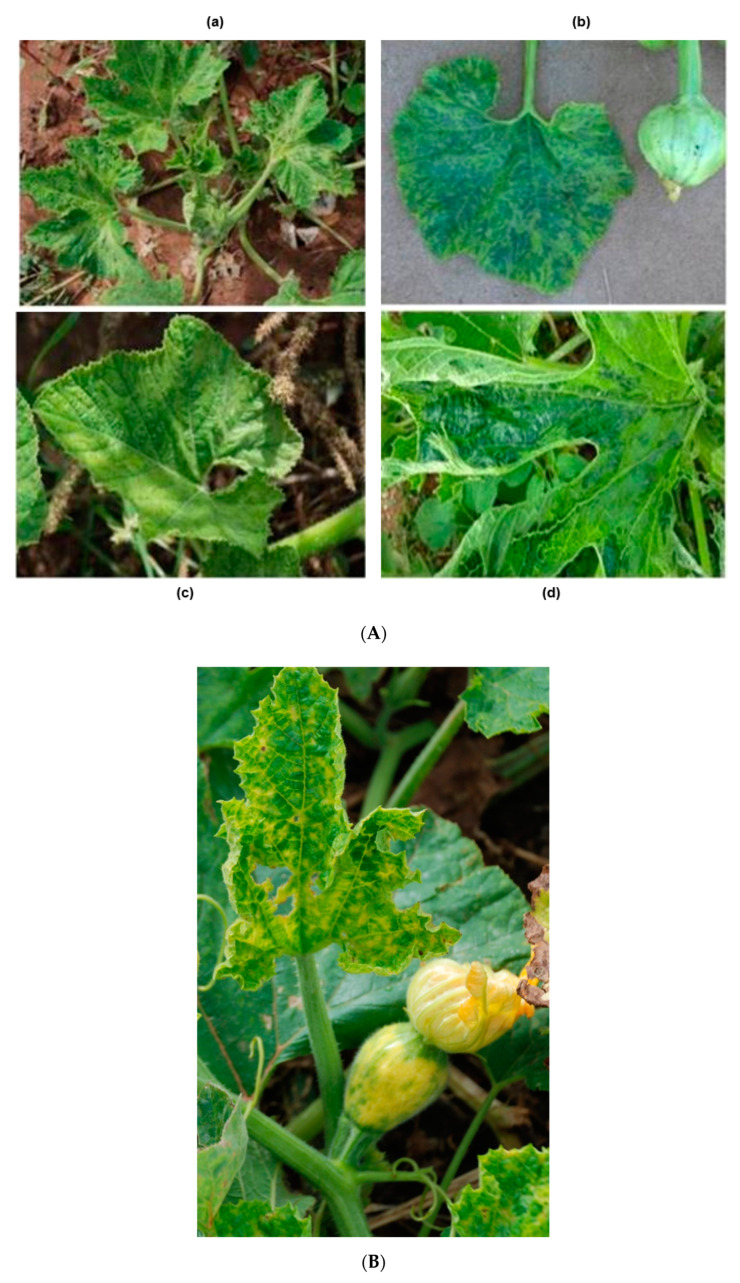
Symptoms observed in infected plants’ leaves and fruits. (**A**) The most common symptoms observed in the leaves of infected cucurbit plants. (**a**) Leaf distortion caused by mixed infection of watermelon mosaic virus (WMV) and papaya ringspot virus W strain (PRSV-W) on squash leaves. (**b**) Mosaic pattern on pumpkin leaves caused by WMV. (**c**) Yellowing in pumpkin caused by ZYMV. (**d**) Mottling and leaf deformation, caused by PRSV-W, on pumpkin leaves. (**B**) Ring spots on pumpkin fruit caused by PRSV-W and yellowing on the leaves by mixed infection of PRSV-W and ZYMV. (**C**) Chlorosis, mosaics, and vein thickening caused by CABYV BL-4 isolate on pumpkin leaves. The pumpkin plant was also infected by PRSV-W.

**Figure 3 pathogens-10-00053-f003:**

Analysis of reverse transcription polymerase chain reaction (RT-PCR) products of three potyviruses by 1% agarose gel electrophoresis. (**A**) Amplification from PRSVCPF and PRSVCPR primers showing DNA band of 963 bp. Lane 1: 10 kb DNA ladder; Lane 2: positive control; Lane 3: negative control; Lane 4–8: PRSV-W dot-immunobinding assay (DIBA)-positive samples; Lane 9–10: PRSV-W DIBA-negative samples. (**B**) Amplification from WMVCPF and WMVCPR primers showing DNA band of 979 bp. Lane 1: 10 kb DNA ladder; Lane 2: positive control; Lane 3: negative control; Lane 4–8: WMV DIBA-positive samples; Lane 9–10: WMV DIBA-negative samples. (**C**) Amplification from ZYMVCPF and ZYMVCPR primers showing DNA band of 902 bp. Lane 1: 10 kb DNA ladder; Lane 2: positive control; Lane 3: negative control; Lane 4–8: ZYMV DIBA-positive samples; Lane 9–10: ZYMV DIBA-negative samples.

**Figure 4 pathogens-10-00053-f004:**
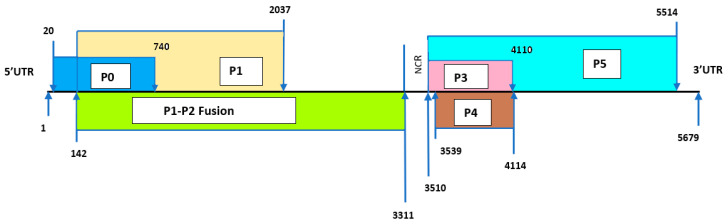
Schematic representation of the genome organization of CABYV BL-4 isolate. The arrows point to the approximate start and end positions of the nucleotides in the particular protein. P1 protein also contains part of the P0 protein region, and P5 protein contains all the P3 protein. The diagram is not drawn to scale.

**Figure 5 pathogens-10-00053-f005:**
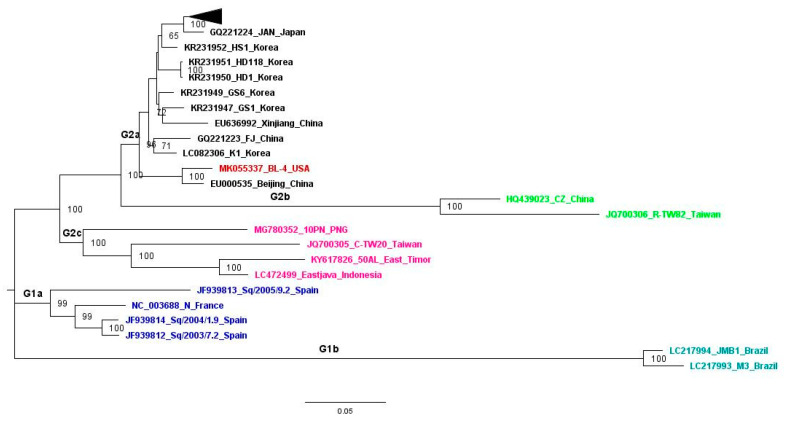
Maximum likelihood (ML) phylogenetic tree deduced in Mega7 using general time reversible (GTR) model. The ML tree is based on 45 complete genome sequences of CABYV isolates available in the NCBI database. GenBank accession number, isolate name, and country of origin are shown on each node. The tree was visualized in Figtree version 1.4.3. The bootstrap values >50 are shown at the respective nodes. The collapsed 22 South Korean isolates are denoted by a dark triangle. The phylogenetic grouping is shown on their respective branches. The US isolate is shown in red color. Isolates from phylogroup G1a are shown in blue, G1b in Torquoise, G2a (except BL-4) in black, G2b in green, and G2c in pink color.

**Figure 6 pathogens-10-00053-f006:**
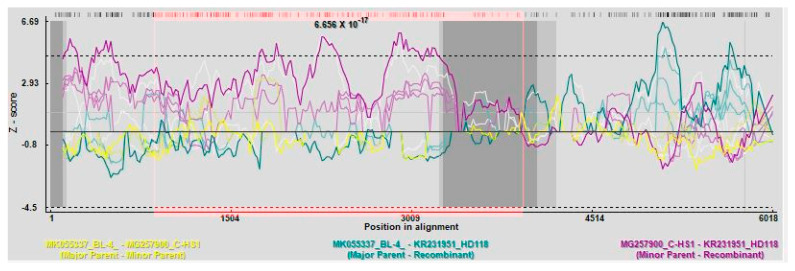
Schematic representation of Recombination Detection Program (RDP) pairwise plot and possible recombination breakpoints in isolate HD118, where the BL-4 isolate from the US serves as a major parent, and C-HS1 from S. Korea serves as a minor parent. The pink area indicates the recombination region.

**Table 1 pathogens-10-00053-t001:** Distribution of viruses in 11 counties of Oklahoma.

County	Atoka	Blaine	Caddo	Carter	Cherokee	Cimmaron	McCurtain	Muskoghee	Payne	Pontotoc	Tulsa	Total
No. samaples	17	309	77	15	27	104	191	267	67	48	209	1331
CMV	0 (0)	0 (0)	0 (0)	0 (0)	0 (0)	0 (0)	0 (0)	0 (0)	0 (0)	0 (0)	0(0)	0 (0)
CABYV	0 (0)	8 (2.6)	0 (0)	0 (0)	0 (0)	0 (0)	0 (0)	6 (2.2)	3 (4.4)	0 (0)	0(0)	17 (1.3)
CGMMV	0 (0)	0 (0)	0 (0)	0 (0)	0 (0)	15 (14.4)	0 (0)	0 (0)	0 (0)	0 (0)	0(0)	15 (1.1)
CYSDV	0 (0)	0 (0)	0 (0)	0 (0)	0 (0)	0 (0)	0 (0)	0 (0)	0 (0)	0 (0)	0(0)	0 (0)
MNSV	0 (0)	0 (0)	0 (0)	0 (0)	2 (7.4)	3 (2.9)	0 (0)	29 (10.9)	0 (0)	24 (50)	18(8.6)	76 (5.7)
PRSV-W	**2 (11.8)**	**171 (55.3)**	29 (37.7)	**6 (40)**	**18 (66.7)**	20 (19.2)	**99 (51.8)**	**219 (82.0)**	**14 (20.9)**	**48 (100)**	**160 (76.6)**	**786 (59.1)**
SLCV	0 (0)	0 (0)	0 (0)	0 (0)	0 (0)	0 (0)	0 (0)	0 (0)	0 (0)	0 (0)	0(0)	0 (0)
SqMV	0 (0)	0 (0)	5 (6.5)	0 (0)	0 (0)	0 (0)	0 (0)	4 (1.5)	0 (0)	0 (0)	10 (4.8)	19 (1.4)
WMV	0 (0)	74 (24.0)	**47 (61.0)**	1 (6.67)	0 (0)	**67 (64.4)**	37 (19.4)	2 (0.7)	0 (0)	0 (0)	47 (22.5)	275 (20.7)
ZYMV	0 (0)	144 (46.60)	6 (7.8)	2 (13.3)	9 (33.3)	41 (39.4)	31 (16.2)	46 (17.2)	5 (7.5)	37 (77.1)	46 (22.0)	367 (27.6)
**Incidence**	2	257	62	7	19	85	124	220	22	48	182	1023
**Percentage**	(11.8)	(83.2)	(80.5)	(46.7)	(70.4)	(81.7)	(64.9)	(82.4)	(32.4)	(100)	(87.1)	(76.9)

Note: The data combine the samples collected from all three years. The values inside the brackets are respective percentages. The viruses with the highest percent incidence within the respective counties are shown in bold script. The sum of individual incidence is more than the total virus incidence due to mixed infections.

**Table 2 pathogens-10-00053-t002:** Incidence of viruses in different hosts.

Host Total Samples	Cantaloupe 67	Cucumber 51	Gourd 3	Pumpkin 745	Squash 156	Watermelon 309	Total 1331
CMV	0 (0)^a^	0 (0)	0 (0)	0 (0)	0 (0)	0 (0)	0 (0)
CABYV	0 (0)	0 (0)	0 (0)	17 (22.9)	0 (0)	0 (0)	17 (1.3)
CGMMV	0 (0)	15 (29.4)	0 (0)	0 (0)	0 (0)	0 (0)	15 (1.1)
CYSDV	0 (0)	0 (0)	0 (0)	0 (0)	0 (0)	0 (0)	0 (0)
MNSV	50 (74.6)	20 (39.2)	0 (0)	1 (0.1)	4 (2.6)	1 (0.3)	76 (5.7)
PRSV-W	**59 (88.0)**	**24 (47.1)**	0 (0)	**450 (60.4)**	**120 (76.9)**	**133 (43.0)**	**786 (59.1)**
SLCV	0 (0)	0 (0)	0 (0)	0 (0)	0 (0)	0 (0)	0 (0)
SqMV	0 (0)	0 (0)	0 (0)	19 (2.6)	0 (0)	0 (0)	19 (1.4)
WMV	6 (9.0)	4 (7.8)	0 (0)	203 (27.2)	31 (19.9)	31 (10.0)	275 (20.7)
ZYMV	29 (43.3)	20 (39.2)	0 (0)	239 (32.1)	57 (36.5)	22 (7.1)	367 (27.6)
Virus incidence	61 (91.0)	39 (76.5)	0 (0)	643 (86.3)	131 (84.0)	149 (48.2)	1023 (76.9)

Note: The values inside the parentheses are their respective percentages. The virus with the highest percent incidence within the respective host is shown in bold script. The sum of individual incidence is more than the total virus incidence due to mixed infections.

**Table 3 pathogens-10-00053-t003:** Incidence of viruses during 2016–2018 growing season.

Virus/Year Total Samples	2016 433	2017 298	2018 600	Total 1331
CMV	0 (0)	0 (0)	0 (0)	0 (0)
CABYV	0 (0)	11 (3.7)	6(1.0)	17 (1.3)
CGMMV	15 (3.5)	0 (0)	0 (0)	15 (1.1)
CYSDV	0 (0)	0 (0)	0 (0)	0 (0)
MNSV	0 (0)	0 (0)	75 (12.5)	75 (5.7)
PRSV-W	**182 (42.0)**	**155 (52.0)**	**449 (74.8)**	**786 (59.1)**
SLCV	0 (0)	0 (0)	0 (0)	0 (0)
SqMV	15 (3.5)	0 (0)	4 (0.7)	19 (1.4)
WMV	173 (40.0)	55 (18.5)	47 ((7.8)	275 (20.7)
ZYMV	54 (12.5)	24 (8.1)	289 (48.2)	367 (27.6)
Virus incidence	340 (78.5)	193 (64.8)	490 (81.7)	1023 (76.9)

Note: The values inside the brackets are respective percentages. The virus with the highest percent incidence within the respective year is shown in bold script. The sum of individual incidence is more than the total virus incidence due to mixed infections.

**Table 4 pathogens-10-00053-t004:** Incidence of viruses tested in different months of the growing season.

Virus/Month Total samples	July 162	August 456	September 465	October 248	Total 1331
CMV	0 (0)	0 (0)	0 (0)	0 (0)	0 (0)
CABYV	0 (0)	6 (1.3)	11 (2.4)	0 (0)	17 (1.3)
CGMMV	0 (0)	15 (3.3)	0 (0)	0 (0)	15 (1.1)
CYSDV	0 (0)	0 (0)	0 (0)	0 (0)	0 (0)
MNSV	12 (7.4)	29 (6.4)	35 (7.5)	0 (0)	75 (5.7)
PRSV-W	**105 (64.8)**	**328 (71.9)**	**218 (46.9)**	**135 (54.4)**	**786 (59.1)**
SLCV	0 (0)	0 (0)	0 (0)	0 (0)	0 (0)
SqMV	0 (0)	12 (2.6)	0 (0)	7 (2.8)	19 (1.4)
WMV	1 (0.6)	38 (8.3)	146 (31.4)	90 (36.3)	275 (20.7)
ZYMV	62 (38.3)	92 (20.2)	132 (28.4)	81 (32.7)	367 (27.6)
Virus incidence	108 (66.7)	355 (77.9)	333 (71.6)	227 (91.5)	1023 (76.9)

Note: The values inside the brackets are respective percentages. The virus with the highest percent incidence within the respective month is shown in bold script. The sum of individual incidence is more than the total virus incidence due to mixed infection.

**Table 5 pathogens-10-00053-t005:** Incidence of viruses tested at different growth stages of hosts.

Virus/Growth Stage Total Samples	Pre-Flowering 133	Fruiting 408	Harvesting 790	Total 1331
CMV	0 (0)	0 (0)	0 (0)	0 (0)
CABYV	0 (0)	9 (2.2)	8 (1.0)	17 (1.3)
CGMMV	0 (0)	0 (0)	15 (1.9)	15 (1.1)
CYSDV	0 (0)	0 (0)	0 (0)	0 (0)
MNSV	0 (0)	11 (2.7)	65 (8.2)	75 (5.7)
PRSV-W	**74 (55.6)**	**247 (60.5)**	**465 (58.9)**	**786 (59.1)**
SLCV	0 (0)	0 (0)	0 (0)	0 (0)
SqMV	0 (0)	12 (2.9)	7 (0.9)	19 (1.4)
WMV	0 (0)	33 (8.1)	242 (30.6)	275 (20.7)
ZYMV	27 (20.3)	106 (26.0)	234 (29.6)	367 (27.6)
Total	74 (55.6)	277 (67.9)	672 (85.1)	1023 (76.9)

Note: The values inside the brackets are respective percentages. The virus with the highest percent incidence within the respective growth stage is shown in bold script. The sum of individual incidence is more than the total virus incidence due to mixed infections.

**Table 6 pathogens-10-00053-t006:** Mixed virus infections and their frequencies among 1023 virus-infected samples.

Double Infection		Triple Infection		Quadruple Infection	
Virus Combination	No. of Infected Samples	Virus Combination	No. of Infected Samples	Virus Combination	No. of Infected Samples
CABYV+PRSV	2	CABYV+WMV+PRSV	1	CABYV+PRSV+WMV+ZYMV	2
CABYV+ZYMV	1	CABYV+WMV+ZYMV	2	SqMV+PRSV+WMV+ZYMV	1
CABYV+WMV	1	CABYV+PRSV+ZYMV	5	MNSV+PRSV+WMV+ZYMV	4
SqMV+PRSV	3	SqMV+WMV+PRSV	3		
SqMV+WMV	9	SqMV+PRSV+ZYMV	3		
MNSV+PRSV	30	MNSV+PRSV+ZYMV	39		
MNSV+ZYMV	2	MNSV+PRSV+WMV	1		
PRSV+WMV	63	PRSV+WMV+ZYMV	30		
PRSV+ZYMV	180				
WMV+ZYMV	52				
Total	343/1023	Total	84/1023	Total	7/1023
% infection	33.52		8.21		0.68
Total mixed Infection %	434/1023 (42.42)				

**Table 7 pathogens-10-00053-t007:** Proteins of CABYV BL-4 isolate, position in the genome, and amino acid length.

Protein	Position in Genome	Amino Acid Length
Silencing Suppressor (P0)	21–740	239
Viral Proteinase (P1)	142–2037	631
Fusion Protein (P1–P2)	142–1488, 1488–3311	1056
Coat Protein (P3)	3511–4110	199
Movement Protein (P4)	3539–4114	191
Readthrough protein (P5)	3511–5514	666

**Table 8 pathogens-10-00053-t008:** Percent similarities of the complete genome nucleotides of the CABYV BL-4 isolates sequenced with other isolates in GenBank.

S. N	Isolate, Country	Accession	Query Cover	Nucleotide Identity
1	Beijing, China	EU000535	100%	97%
2	GS6, South Korea	KR231949	100%	95%
3	CABYV-JAN, Japan	GQ221224	100%	94%
4	Sq/2004/1.9, Spain	JF939814	100%	89%
5	N, France	X76931	100%	89%
6	10PN, Papua New Guinea	MG780352	97%	88%
7	JMB1, Brazil	LC217994	61%	88%
8	50Al, East Timor	KY617826	96%	88%
9	Indonesia	LC472499	97%	88%
10	CABYV-C-TW20, Taiwan	JQ700305	100%	87%

**Table 9 pathogens-10-00053-t009:** Putative recombination breakpoints, their positions in the genome genes affected, and genes involved in complete genomes of 45 CABYV isolates available in GenBank. The recombination predicted by at least four methods among seven in RDP4 is shown. The method showing the highest values is in bold script.

Events	Recombinant Isolate	Breakpoints (nt)		Genes Affected	Parental Sequences		RDPs *	Highest *p*-Value
		Beginning	Ending		Major	Minor		
1	CZ	3140	5691	P1–P2, IR, P3, P4, P3–P5, 5′UTR	N	HD-118	**R**GBMCS3	5.87 × 10^−63^
2	Eastjava	1994	3507	P1, P1–P2, IR	50AL	C-TW2	R**G**BMCS3	5.87 × 10^−54^
3	R-TW82	1492	3379	P1, P1–P2, IR	C-TW20	Sq-2004	RGBMC**S**3	1.03 × 10^−39^
4	Sq-2005	4866	5675	P3–P5, 5′UTR	M-BY1	Sq-2003	R**G**BMC-S3	5.97 × 10^−33^
5	K1	711	3402	P0, P1, P1–P2, IR	GM7	NW2	RGBMCS**3**	7.03 × 10^−26^
6	HS1	861	4881	P2, P1–P2, IR, P3, P4, P3–P5	Xinjiang	CY6	RGBMCS**3**	9.83 × 10^−26^
7	GS6	4475	5671	P3–P5, 5′UTR	CY3	Xinjiang	RGBMCS**3**	2.19 × 10^−24^
8	FJ	4891	5682	P3–P5, 5′UTR	CY3	HD118	RGBMCS**3**	7.14 × 10^−17^
9	Xinjiang	1234	3433	P1, P1–P2, IR	HD118	M-CY3	RGBMCS**3**	3.22 × 10^−16^
10	CZ	251	744	P0, P1–P2	10PN	C-AS1	R**G**BMCS3	6.09 × 10^−14^
11	GS1	4301	5682	P3–P5, 5′UTR	GM16	HS2	RGBMCS**3**	1.06 × 10^−19^
12	HD118, HD1	854	3862	P1, P1–P2, IR, P3, P4, P3–P5	BL-4	C-HS1	RBMC**S**3	1.32 × 10^−16^
13	C-TW2	5447	5600	3′UTR	Sq-2003	BL-4	**R**GBC	4.40 × 10^−7^
14	10PN	5515	5607	3′UTR	CY3	N	**R**GBMCS	6.23 × 10^−8^
15	HS1, Xinxiang	1	719	5′UTR, P0, P1	K1	GS2	**R**GBMCS3	3.07 × 10^−9^
16	GS6	1665	4160	P1, P1–P2, IR, P3, P4, P3–P5	GS2	CY3	RGBMCS**3**	6.07 × 10^−9^
17	FJ	1	636	5′UTR, P0, P1	CY4	K1	**R**GBMC3	4.6 × 10^−8^
18	WM-YS10	1637	3589	P1, P1–P2, IR, P3, P4, P3–P5	CY3	C-AS1	RBMC**S**3	1.63 × 10^−9^
19	SW2, SW1, CY3	1159	1665	P1, P1–P2	C-HS1	NW18	RG**B**MCS3	8.03 × 10^−5^
20	JAN	1182	3206	P1, P1–P2	GM7	GM16	M**C**S3	1.74 × 10^−5^
21	NW2, NW1, NW18	1235	2730	P1, P1–P2	M-BY1	CY-6	RBM**C**S3	3.39 × 10^−4^
22	NW5, CY4	1666	4469	P1, P1–P2, IR, P3, P4, P3–P5	M-BY1	C-AS1	BMC**S**3	7.54 × 10^−7^

* R = RDP, G = GENECOV, B = BootScan, M = MaxChi, C = Chimaera, S = SiScan, 3 = 3Seq.

**Table 10 pathogens-10-00053-t010:** List of viruses tested by DIBA. The modes of transmission and their respective genera and families are also provided in the table.

Virus-Tested	Abbreviation	Transmission	Genus	Family
*Cucumber green mottle mosaic virus*	CGMMV	Seed/fungus	*Tobamovirus*	*Vigaviridae*
*Cucumber mosaic virus*	CMV	Aphids	*Cucumovirus*	*Bromoviridae*
*Cucurbit aphid-borne yellows virus*	CABYV	Aphids	*Polerovirus*	*Luteoviridae*
*Cucurbit yellow stunting disorder virus*	CYSDV	Whitefly	*Crinivirus*	*Closteroviridae*
*Melon necrotic spot virus*	MNSV	Seed/fungus	*Carmovirus*	*Tombusviridae*
*Papaya ringspot virus*	PRSV	Aphids	*Potyvirus*	*Potyviridae*
*Squash leaf curl virus*	SLCV	Whitefly	*Begomovirus*	*Geminiviridae*
*Squash mosaic virus*	SqMV	Beetle	*Comovirus*	*Secoviridae*
*Watermelon mosaic virus*	WMV	Aphids	*Potyvirus*	*Potyviridae*
*Zucchini yellow mosaic virus*	ZYMV	Aphids/seed	*Potyvirus*	*Potyviridae*

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
