# Peer review of "High Prevalence of Three Potyviruses Infecting Cucurbits in Oklahoma and Phylogenetic Analysis of Cucurbit Aphid-Borne Yellows Virus Isolated from Pumpkins"

_pathogens, 2021, doi:10.3390/pathogens10010053_

Round 1

Reviewer 1 Report

This manuscript aims to extend the existing knowledge on the viral diseases affecting cucurbit crops in Oklahoma. It describes the relative incidence and distribution of viruses, in addition to the identification and genetic characterization of a new CABYV species.

This work is interesting and has merit, whereas there is a large monitoring and identifying a broad spectrum of viruses in those symptomatic samples, with the occurrence for some of them either in mixed infections, or recently and scarcely reported, and even CABYV detection for first time in Oklahoma.

Based on the data presented, the authors describe the incidence of viruses among cultivated cucurbit plant species. However, there is a key weakness with the manuscript in its current form. The authors invoke that sampling was addressed to leaves showing symptoms (line 371) to estimate incidence, and there is an unclear monitoring strategy (including healthy plants) to this end, as incidence is defined by the occurrence of new diseased-plants from the total of plants sampled over time at one place. This aspect should be considered and, at least, a re-interpretation of the term. Moreover, distribution of viruses in different counties, hosts, seasons and growth stages of plants is nice and worthwhile. While it is appropriate, I think that is quite hard to understand the viral dynamics over each factor without following those analysis on the same counties, hosts or plants at different stages. A random factor could be introducing bias, and in turn, a lack of power in the statistical analysis. Therefore, any potential effect associated to those factors may be as a consequence of other factors influencing their dynamics.  In fact, I would recommend to amend all that results providing proportions and statistic data in only one section considering previous comments. This combination would also condensate the tables and will make it easier to follow.  Overall, the work is worthwhile but I think that the revision level for this work is major, and would need to edit substantial and appropriate changes into text sections for a broad virologist reader.   

I would also recommend to the authors to address couple of minor things.

Line 37: it is 50% of emerging viral diseases.

Fig1. Possibly, it fits better as supplementary material rather than in the main text.

…. (

Author Response

Reviewer#1

Please see the attached responses from authors to your comments.

thanks

Akhtar

Reviewer 2 Report

The authors characterized the incidence of different virus in difference agricultural districts of Oklahoma. They analyzed viruses spreading among different hosts and their incidence rates among different season and year. Overall, the data are well analyzed and presented. However, the conclusion and discussion are not well written and they are just description of the results parts. I strongly suggest the authors rewrite the discussion part. There are some other points I would like the author to address.

  1. Part 2.2: Please give a discussion why the viral incidence rates vary significantly among difference counties?
  2. Part 2.3: The viral incidence rates are different among different hosts. Please test whether the virus has preference to infect some hosts, such as Cantaloupe and pumpkin.
  3. Part 2.4: The viral incidence rates vary significantly among the three years analyzed. Are there any climate changes that cause the variation of incidence rates? What the data here can demonstrate?
  4. Part 2.8: Please provide the PCR results for the identifications of the viruses.
  5. Are there any variation of infection rates for specific hosts, such as Cantaloupe and pumpkin among different years?
  6. Please give a discussion on the significance of this study. Are there any implications that this study can provide on the prevention of these viruses spreading?

Author Response

Reviewer#2

Please see the attached responses from authors to your comments.

thanks

Akhtar

Reviewer 3 Report

Dear colleagues,

This review concerns “High incidence of three potyviruses infecting cucurbits in Oklahoma and phylogenetic analysis of Cucurbit aphid-borne yellows virus isolated from Pumpkin”, by Vivek Khanal, Harrington Wells and Akhtar Ali. As detailed experiments I recommend it for an international audience with this journal. However, several points have to be precised and a major revision is requested. Please notice that the three major points of my comments (at the beginning) are very important (mandatory…) for a suitable value and understanding of the article.

I deeply hope to see this article published in this journal,

The three major points are :

  • Actually this topics being of real interest, the references have to be updated as in the present form this article needs to be improved and sustained by much more recent articles: moreover these articles have to be used efficiently in the discussion in all their relevant details (observations, values, comparisons…) for the present purpose, much more than just citing them; see also the references cited in all these papers and extract the relevant data. For instance, checking in the “web of science” by the key-words used in the title and abstract (and acronyms of viruses), quite many research papers and/or review(s) from the very recent years (months…) seem to be relevant, see just below among the last ones:

-For taxonomy and phylogeny of Cucurbitaceae:

Phylotranscriptomics in Cucurbitaceae Reveal Multiple Whole-Genome Duplications and Key Morphological and Molecular Innovations By: Guo, Jing; Xu, Weibin; Hu, Yi; et al.

MOLECULAR PLANT  Volume: ‏ 13   Issue: ‏ 8   Pages: ‏ 1117-1133    Published: ‏ AUG 3 2020

Gourds and Tendrils of Cucurbitaceae: How Their Shape Diversity, Molecular and Morphological Novelties Evolved via Whole-Genome Duplications By: Barrera-Redondo, Josue; Lira-Saade, Rafael; Eguiarte, Luis E. MOLECULAR PLANT  Volume: ‏ 13   Issue: ‏ 8   Pages: ‏ 1108-1110    Published: ‏ AUG 3 2020

Phylotranscriptomics in Cucurbitaceae Reveal Multiple Whole-Genome Duplications and Key Morphological and Molecular Innovations By: Guo, Jing; Xu, Weibin; Hu, Yi; et al.

MOLECULAR PLANT  Volume: ‏ 13   Issue: ‏ 8   Pages: ‏ 1117-1133     Published: ‏ AUG 3 2020

-For WMV virus or other viruses in other areas and countries:

Molecular variability of watermelon mosaic virus isolates from Argentina By: Pozzi, E.; Perotto, M. C.; Bertin, S.; et al. EUROPEAN JOURNAL OF PLANT PATHOLOGY  Volume: ‏ 156   Issue: ‏ 4   Pages: ‏ 1091-1099   Published: ‏ APR 2020

Current trends in protected cultivation in Mediterranean climates

By: Fernandez, J. A.; Orsini, F.; Baeza, E.; et al. EUROPEAN JOURNAL OF HORTICULTURAL SCIENCE  Volume: ‏ 83   Issue: ‏ 5   Pages: ‏ 294-305   Published: ‏ OCT 2018

Molecular diversity of main cucurbit viruses in Syria By: Chikh-Ali, Mohamad; Natsuaki, Tomohide; Karasev, Alexander, V JOURNAL OF PLANT PATHOLOGY  Volume: ‏ 101   Issue: ‏ 4   Pages: ‏ 1067-1075   Published: ‏ NOV 2019

A review of the plant virus and viroid records for Tasmania By: Guy, P. L.; Cross, P. A.; Wilson, C. R. AUSTRALASIAN PLANT PATHOLOGY  Volume: ‏ 49   Issue: ‏ 5   Pages: ‏ 479-492   Published: ‏ SEP 2020 10.1007/s13313-020-00725-5JUN 2020

Distribution and evolution of the major viruses infecting cucurbitaceous and solanaceous crops in the French Mediterranean area By: Desbiez, Cecile; Wipf-Scheibel, Catherine; Millot, Pauline; et al. VIRUS RESEARCH  Volume: ‏ 286      Published: ‏ SEP 2020

-For RNA viruses:

Long Noncoding RNAs in Plant Viroids and Viruses: A Review By: Shrestha, Nipin; Bujarski, Jozef J. PATHOGENS  Volume: ‏ 9   Issue: ‏ 9     Article Number: 765   Published: ‏ SEP 2020

-For potyviruses:

Application of Reverse Genetics in Functional Genomics of Potyvirus By: Kannan, Maathavi; Zainal, Zamri; Ismail, Ismanizan; et al. VIRUSES-BASEL  Volume: ‏ 12   Issue: ‏ Published: ‏ AUG 2020

The Potyviruses: An Evolutionary Synthesis Is Emerging By: Gibbs, Adrian J.; Hajizadeh, Mohammad; Ohshima, Kazusato; et al.VIRUSES-BASEL  Volume: ‏ 12   Issue: ‏ 2     Article Number: 132   Published: ‏ FEB 2020

2 the second point concerns taxonomy sensu lato. In order to make this article more readable for an international audience, for all plants cited please use at least once their latin names (in italics or not, for instance for the family level  (Cucurbitaceae) italics are not relevant, please check all these details in academic international taxonomy sites, as for instance the  International Code of Nomenclature for algae, fungi, and plants); check if for each taxon (experimented in each county) it refers actually to the same species, subspecies, variety etc (there may be different origins to the populations, apparently coming from agronomy selection as this is very usual for all cultivated plants, please use these data eventually in the discussion for the variability or not of the features observed);

3 the third point concerns morpho-anatomical changes apparently caused by viruses. In this respect:

  • the caption of Figure 2 is not clear at all: please precise which viruses are (potentially) present for each leaf (and which taxa are concerned, as detailed in your tables 1 and 2);
  • In the discussion part (paragraph 312-327), the effects of viruses is too weakly sustained by references;
  • Moreover, sections of leaves would be extremely welcome to demonstrate and precise the effect of the virus (es) on the different types of cells of the leaves (epidermis, inner cells, xylem or phloem, veins…), this can be very easily done and would bring a real increase in the scientific interest of this article. In order to increase greatly this part of the paper, check in the references of these viruses which is the effect of each virus (and on which taxa as it is usually very related), this could lead to bring a diagnostic very usefull in agronomy. Data on the level of concentration of these viruses in the plants experimented would be welcome (causing changes or not in the leaves), but I understand that this latter remark concerns probably another study and article…

The minor points are:

1 for Figure 4 more colours would be suitable to understand more rapidly the different groups of viruses as discussed in the paper;

2 for 4.1, precise the number of leaves experimented;

3 in table S1 of the supplementary data, put italics for latin names of plants (genus, species… levels) in column 3; check also  if for viruses (column 1; see for instance  ICTV Code The International Code of Virus Classification and Nomenclature October 2018) the Family level is in italics or not (it is not for plants…).

Author Response

Reviewer#3

Please see the attached responses from authors to your comments.

thanks

Akhtar

Round 2

Reviewer 1 Report

I'm pleased to see the improvements in this revised manuscript.

I see no major issues, although there is one minor methodological thing. This is about the performance of the DIBA. I would recommend to extent a description, even if previously described, about how was the procedure.

Author Response

See the attached responses.

Reviewer 2 Report

The authors have addressed all of my concerns in the revised manuscript.

Author Response

See the attached responses.

Reviewer 3 Report

Good afternoon,

In my opinion the whole article is now well-done, however please check again the below paragraph in the introduction as for plants, family taxonomical level requires no italics (Cucurbitaceae) while genus level requires italics  (Cucurbita). Another point is that it is not relevant to write "the family Cucurbitaceae" as it is redondant words (the word "Cucurbitaceae" meaning the family taxonomical level it has not to be used with the word "family". A better sentence is "the members of  Cucurbitaceae, family commonly..." (separate family and Cucurbitaceae). In the same way, two lines after, delete family and just use "The Cucurbitaceae includes...").

Have a nice end of year,

The members of the family Cucurbitaceae, commonly referred to as cucurbits, are major cash
28 crops in the United States (U.S.) and worldwide. The Cucurbitaceae family includes 15 tribes, 95-97
29 genera and approximately 1000 species, which are grown in tropical and sub-tropical regions [1, 2,
30 3]. Among them, the genus Cucumis (including cucumber and melon), Citrullus (watermelon), and
31 Cucurbita (pumpkin and squash) are more popular. Archeological evidence suggests that Pumpkins
32 and squashes (Cucurbita spp)

Author Response

Thank the reviewer for the comments. Please see the revised manuscript.